# Self-Aligned Thin-Film Patterning by Area-Selective Etching of Polymers

Chao Zhang , Markku Leskelä and Mikko Ritala *

Department of Chemistry, University of Helsinki, 00014 Helsinki, Finland; chao.zhang@helsinki.fi (C.Z.); markku.leskela@helsinki.fi (M.L.)
* Correspondence: mikko.ritala@helsinki.fi; Tel.: +358-2941-50193

**Abstract:** Patterning of thin films with lithography techniques for making semiconductor devices has been facing increasing difficulties with feature sizes shrinking to the sub-10 nm range, and alternatives have been actively sought from area-selective thin film deposition processes. Here, an entirely new method is introduced to self-aligned thin-film patterning: area-selective gas-phase etching of polymers. The etching reactions are selective to the materials underneath the polymers. Either $O_2$ or $H_2$ can be used as an etchant gas. After diffusing through the polymer film to the catalytic surfaces, the etchant gas molecules are dissociated into their respective atoms, which then readily react with the polymer, etching it away. On noncatalytic surfaces, the polymer film remains. For example, polyimide and poly(methyl methacrylate) (PMMA) were selectively oxidatively removed at 300 °C from Pt and Ru, while on $SiO_2$ they stayed. $CeO_2$ also showed a clear catalytic effect for the oxidative removal of PMMA. In $H_2$, the most active surfaces catalysing the hydrogenolysis of PMMA were Cu and Ti. The area-selective etching of polyimide from Pt was followed by area-selective atomic layer deposition of iridium using the patterned polymer as a growth-inhibiting layer on $SiO_2$, eventually resulting in dual side-by-side self-aligned formation of metal-on-metal and insulator (polymer)-on-insulator. This demonstrates that when innovatively combined with area-selective thin film deposition and, for example, lift-off patterning processes, self-aligned etching processes will open entirely new possibilities for the fabrication of the most advanced and challenging semiconductor devices.

**Keywords:** self-aligned thin-film patterning; area selective etching of polymers; area selective atomic layer deposition





## 1. Introduction

Development of integrated circuits (IC) along Moore's law has brought the technology to a level where the smallest features of semiconductor devices are smaller than 10 nm. This causes major challenges for techniques used to pattern such structures. For example, the most complicated microprocessors consist of fin-shaped transistors that are interconnected with over 15 layers of metal wires and the same number of metal vias in between. Each layer is deposited and patterned on its own and must be carefully aligned with the underlying layer. While significant progress has been made in resolution with multiple patterning techniques and extreme ultraviolet lithography, several masks and lithography cycles are needed for each layer, and alignment errors between subsequent patterning steps increase significantly as the device structures get smaller [1,2]. Lithography tools are the most expensive ones in semiconductor fabrication yet are prone to costly errors. Therefore, there is an urgent need to replace, or at least, simplify as many lithography process steps as possible with self-aligning thin film fabrication processes, especially at the smallest and most critical dimensions for alignment. So far, this has been attempted and realized with varying levels of success by area-selective thin film deposition methods, such as atomic layer deposition (ALD) and chemical vapor deposition (CVD) [3–8]. In these processes, the

self-alignment arises from the selectivity of the chemistry so that the film-forming reactions proceed only on the growth areas and not on the nongrowth areas. These areas consist of different materials that are either inherently present on the surface of the IC device being fabricated (semiconductors, insulators, metals) or created by modifying the surfaces with proper passivating (self-assembled monolayers, polymer films) or activating agents (e.g., noble metals) [1,9]. The inherent area selectivity would be the ideal solution for the self-aligned film patterning because no additional process steps are needed [1]. While some processes do show robust selectivity, a common observation is that when the film on the growth areas reaches a certain thickness, growth also begins on the nongrowth areas, causing a loss of the selectivity [10–12]. Corrective etch-back processes have been developed to mitigate this problem [13,14].

In this paper, we introduce a new approach that may bring a paradigm shift to self-aligned thin-film patterning: area-selective etching of polymers. These gas-phase etching processes are inherently selective to the materials under the polymers so that the polymers are removed only from the surfaces of certain materials and remain on others. These differences arise from the catalytic effects of the underlying materials on the polymer removal reactions. Another key requirement and feature of these etching processes is that the etchant molecules must diffuse through the polymer layer and become activated on the surface of the underlying catalytic material. As is well known, polymers have high permeation for small molecules and therefore require diffusion barriers in packaging [15] and electronic applications [16,17]. Here, an advantage is taken from the high permeation.

Figure 1 shows the key aspects of self-aligned polymer etching. The starting surface is a patterned surface consisting of catalytic metal regions (Pt in this case) and noncatalytic $SiO_2$ regions. Such a structure exists there because of the device being fabricated, so at least one lithography step must precede the new approach. Therefore, self-aligned etching cannot replace the very first lithography step but potentially many of the following ones. A continuous polymer film is first deposited on the patterned surface by any applicable method: spin-coating, dip-coating, CVD, or MLD (molecular layer deposition). The sample is then annealed in the presence of molecular oxygen at a properly selected temperature (one at which $O_2$ does not react with the polymer to any significant extent without a catalyst, and therefore, the polymer remains intact on $SiO_2$). By contrast, on the catalytic Pt surface, the $O_2$ molecules diffusing through the polymer are dissociated into atomic oxygen, which readily reacts with the polymer, combusting it away in the form of small molecules such as carbon oxides and water. Once the reaction is completed, the resulting surface consists of the bare Pt and the polymer remaining on the $SiO_2$ surface. If so desired, the process can be continued with area-selective deposition of metal on Pt with the patterned polymer as a growth-inhibiting layer on $SiO_2$, eventually resulting in the highly desired dual side-by-side self-aligned formation of metal-on-metal and insulator (polymer)-on-insulator (Figure 1) [1]. If the polymer is not needed in the device, it can be removed after the metal deposition. Alternatively, the patterned polymer can be exploited in a lift-off process by using less conformal deposition methods such as sputtering or evaporation (Figure 1). Lift-off patterning can also be performed with only partially selective ALD processes, provided that the film growth and coverage on the polymer are poor enough to allow its subsequent etching [18,19]. Finally, it has been noted that when applied as shown in Figure 1, the polymer and the subsequently deposited film, if any, exactly copy the original pattern. If this is not desired, selected areas must be covered by a blocking layer or resist either before or after the self-aligned polymer etching process. While this adds new process steps, patterning of the additional layer does not need as high resolution or alignment accuracy as conventional photolithography. In other words, as with area-selective deposition processes, self-aligned etching cannot always eliminate lithography completely in a given step, but it can relax the resolution requirements and eliminate edge placement errors.

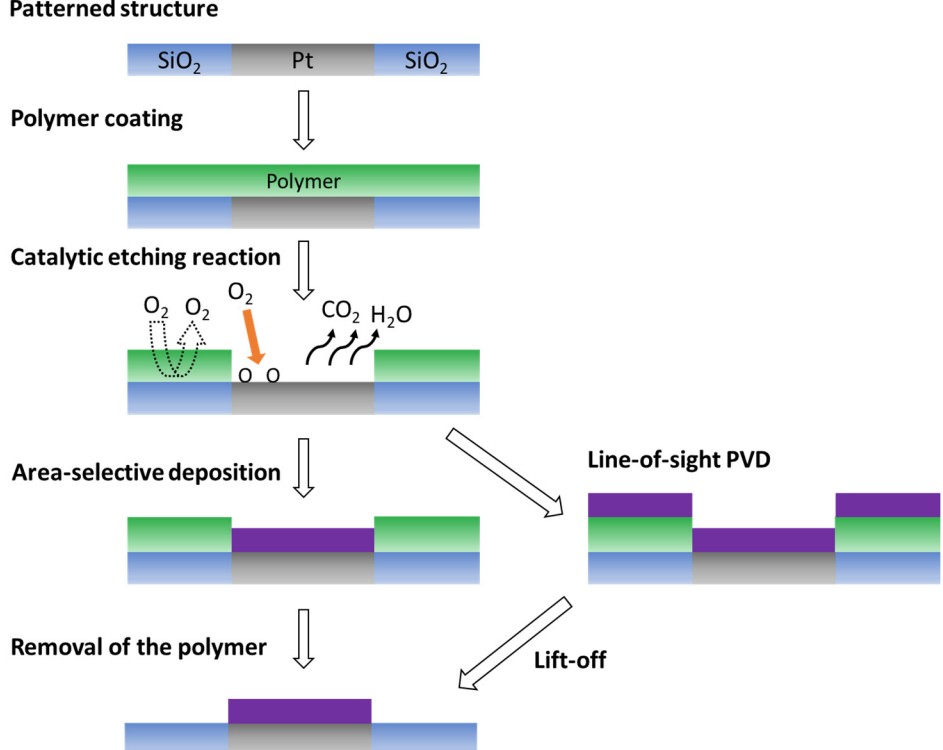

**Figure 1.** Schematics showing self-aligned polymer etching and the subsequent film patterning through area-selective deposition and lift-off processes.

We further demonstrate that the etchant molecule can be $H_2$ as well as $O_2$. Furthermore, in some cases, no etchant molecules are needed, as area-selective etching occurs only by catalytic cracking of the polymer. These alternatives give freedom in choosing between oxidative and reducing etching conditions. Finally, and very importantly, not only metals but also dielectrics can serve as the catalytic surface.

Novel self-aligned etching processes offer a very simple and cheap alternative for film patterning. No expensive lithography tools are needed, and exposure and development steps are eliminated, as compared to the common lithography processes. Therefore, the new approach may find use not only at the smallest dimensions with the most stringent alignment requirements, but also on all other levels of patterning. It is therefore expected that in the hands of process and device design engineers, self-aligned etching processes, cleverly combined with the already existing self-aligned thin film deposition processes, will open entirely new possibilities, a paradigm shift, for the fabrication of the most advanced and challenging IC devices by allowing the elimination of a good number of lithography processing steps. Applications beyond semiconductor devices are also likely.

## 2. Materials and Methods

Polyimide thin film preparation. Polyimide thin films were prepared by MLD in a hot-wall, crossflow F120 reactor (ASM Microchemistry). The operating pressure of the reaction chamber was around 10 mbar. Inert nitrogen gas with a purity of 99.999% was used as the carrier and purge gas. 1,6-diaminohexane (DAH) and pyromellitic dianhydride (PMDA), purchased from Sigma-Aldrich, were used as the precursors for the polyimide MLD process [20]. The evaporation temperatures of DAH and PMDA were 45 and 165 °C, respectively. The pulse times of 2 s and 5 s were used for DAH and PMDA, respectively. The purge time was 3 s for DAH and 5 s for PMDA. Si wafers of size $5 \times 5$ cm$^2$ covered with different top layers, i.e., Pt, Ru, Cu, Co, W, Ti, CeO$_2$, or native SiO$_2$, were used as the substrates for polyimide thin film growth. Thin films of Pt and Ti with thicknesses of around 60 and 50 nm, respectively, were prepared by electron beam evaporation, and the film

thicknesses were measured in situ by a quartz crystal microbalance (QCM, Intellemetrics, Scotland, UK). Thin films of Ru and $CeO_2$ were deposited by ALD using native $SiO_2$-terminated Si as substrates. Ru film with a thickness of 40 nm was grown from $RuCp_2$ and $O_2$ at 300 °C [21]. $CeO_2$ film with a thickness of 40 nm was deposited from $Ce(thd)_4$ and $O_3$ at 250 °C [22]. Si wafers coated with Cu, Co, or W were received from ASM Microchemistry. They had structures of 1 μm Cu/30 nm Ta/Si, 50 nm Co/100 nm $SiO_2$/Si, and 25 nm W/20 nm Ti/Si, respectively. Unless otherwise noted, the metal films were used as received without any chemical pretreatment.

PMMA thin film preparation. PMMA thin films were made on the different surfaces by spin-coating from a solution of 2% PMMA in toluene. PMMA was purchased from Alfa Aesar and had an average molecular weight of 350,000. After the spin-coating, the samples were baked in an oven under ambient atmosphere at 100 °C for 1 h to remove the toluene solvent. The PMMA film thicknesses were around 80–130 nm, as measured by ellipsometry. The Pt, Ru, $CeO_2$, Cu, Co, W, and Ti thin films and the native $SiO_2$ were the same as those used for the polyimide thin film preparation. In addition, a 60 nm Pd film was prepared by sputtering on an $Al_2O_3$-coated Si substrate.

Preparation of patterned Pt/native $SiO_2$ samples. Two methods were used to prepare patterned Pt/$SiO_2$ samples. The first method was shadow mask evaporation: a ~10 nm thick Ti as an adhesion layer and ~80 nm thick Pt film were evaporated on a native $SiO_2$-terminated Si wafer through a shadow mask. The patterned surface consisted of alternate lines of Pt and $SiO_2$. All the line widths were 50 μm. The second method used was a lift-off technique: a ~100 nm thick PMMA thin film was spin-coated on a Si wafer as described above but by using a 3% PMMA toluene solution. The PMMA thin film was then patterned by photolithography using a shadow mask and UV light at wavelengths of 250 and 190 nm for 15 min, followed by a development in 3:1 isopropanol/water solution for 1 min. The sample was rinsed with isopropanol and water and then blow-dried with nitrogen. The sample was further dried in an oven at 100 °C for 1 h. Next, around 3 nm Ti as an adhesion layer and 20 nm Pt were evaporated on the sample. After the metal evaporation, the sample was immersed in acetone for 5 min to remove the remaining PMMA together with the metal layers evaporated on top, resulting in a patterned Pt/$SiO_2$ surface. The prepared sample was rinsed with acetone and isopropanol and then finally blow-dried with $N_2$.

The patterned Pt/native $SiO_2$ surfaces were uniformly coated with 90 nm thick polyimide thin films. The samples were then annealed in an oven under an ambient atmosphere at 300 °C for 30 min to selectively etch polyimide from the top of the Pt. An Ir ALD process was performed on the annealed sample to demonstrate the feasibility of achieving area-selective ALD of Ir on Pt. Ir films were deposited from $Ir(acac)_3$ and $O_2$ at 250 °C [23].

Film characterization. Thicknesses of the polymer thin films were measured by an FS-1 multiwavelength ellipsometer (Film Sense, Lincoln, NE, USA). Scanning electron microscope (SEM) was used to confirm the success in the area-selective etching of polyimide thin films from Pt versus the native $SiO_2$. A combination of SEM and energy dispersive X-ray spectroscopy was used to confirm the subsequent area-selective ALD of Ir on Pt versus the polyimide-covered native $SiO_2$. A Hitachi S-4800 field emission scanning electron microscope (Hitachi, Tokyo, Japan) equipped with an INCA 350 EDX spectrometer was used for the SEM and EDX characterization.

## 3. Results and Discussion

Fundamentals for the area-selective etching of polymers were studied by depositing polyimide and poly(methyl methacrylate) (PMMA) thin films on various surfaces and measuring how fast their thicknesses decrease on the different materials at different temperatures in different atmospheres. Native $SiO_2$ on silicon was used as the noncatalytic surface in all the experiments. While it is hard to prove that the native $SiO_2$ would not have any catalytic effect on the polymer etching reactions, the etching rates on the native $SiO_2$ were anyhow always the lowest among the surfaces being examined, thereby justifying its assignment as the noncatalytic surface. Noble metals of Pt and Ru as catalytic surfaces were

most extensively studied, while several other metals were tested as well, such as Pd, Cu, Co, W, and Ti. In addition to the metal catalysts, dielectric $CeO_2$ was also studied because $CeO_2$ is a well-known soot combustion catalyst commonly used in diesel engines [24]. Soot combustion catalysts operate with the same principle as proposed here, that is, the underlying catalyst activates $O_2$ molecules to combust the overlaying carbonaceous material. All these materials were in the form of blanket thin films on silicon.

It is noted that $CeO_2$ and some of the metals (Pt, Pd) are not common materials in current semiconductor devices. They were selected for these first tests regardless because of their known catalytic properties. Ideally, one should develop processes that work with materials used in the devices. Alternatively, one can envision adding a thin layer of catalytic material on top of existing materials, e.g., $CeO_2$ on top of another $SiO_2$ based insulator, to enable the polymer etching.

### 3.1. Catalytic Combustion of Polyimide in $O_2$ Atmosphere

As the first self-aligning polymer etching reaction, catalytic combustion of polyimide with $O_2$ was studied. Polyimide thin films with thicknesses around 80–120 nm were deposited by MLD [20] on Pt, Ru, and $CeO_2$ as potential catalytic surfaces and native $SiO_2$ as the noncatalytic surface. The polyimide coated samples were annealed in an oven under ambient atmosphere at temperatures of 170–300 °C for various times. Film thicknesses were measured by ellipsometry.

As seen in Figure 2, up to 250 °C, the polyimide combustion on the native $SiO_2$ was very slow. After the initial shrinkage, which we attribute to a densification of the polymer film, the film thickness stayed constant. At 300 °C, the decrease in thickness of the polyimide film became obvious, marking the onset temperature of the polyimide combustion on $SiO_2$. In the absence of a catalyst, the combustion was still slow; a decrease in thickness of only ~40 nm was measured after 4 h annealing. By contrast, on Pt, the polyimide combustion was much more efficient, with a lower onset temperature and much higher etching rates. The onset temperature on Pt was 250 °C, at which a polyimide film with a thickness of around 120 nm was completely removed after 4 h annealing, while under the same condition, the polyimide thin film on the native $SiO_2$ remained nearly intact. At 300 °C, 10 min annealing was enough to etch away an 85 nm-thick polyimide from the Pt surface, whereas on native $SiO_2$, no noticeable thickness decrease was measured after 10 min annealing (Figure 2). Therefore, a magnificent increase of the polyimide combustion rate was measured on Pt, and consequently, an outstanding selectivity between Pt and native $SiO_2$ surfaces is demonstrated. Obviously, if the original polyimide film thickness were lower, the required etching time would shorten accordingly.

On Ru, 300 °C was the onset temperature at which polyimide films could be combusted away (Figure 2). At this temperature, a 90 nm-thick polyimide thin film was almost completely removed after 1 h annealing. Although the catalytic efficiency of Ru is lower than that of Pt, a good selective removal was still achieved between Ru and native $SiO_2$. The thickness of the polyimide film on the native $SiO_2$ decreased by only around 22 nm, while a roughly 90 nm-thick film was removed from the Ru surface. In other words, starting with a patterned Ru/$SiO_2$ surface with a 90 nm-thick polyimide film uniformly coated on top, after 1 h annealing, a polyimide film around 70 nm thick remained on the native $SiO_2$, while the polyimide was completely removed from the Ru surface, thereby resulting in a self-aligned patterning of the polyimide. In this respect there is a significant difference between the two self-aligned patterning methods: area-selective etching is useful, even if quite far from perfect, whereas area-selective deposition suffers from the smallest deviations from the perfect selectivity, that is, any growth on the nongrowth areas may be detrimental.

Within the examined annealing temperature range, no catalytic effect of $CeO_2$ on the polyimide combustion was observed as compared with $SiO_2$ (Figure 2). On the other hand, the $CeO_2$ surface did exhibit excellent catalytic performance in the combustion of PMMA thin films, as is discussed below. The reason for this difference is currently unknown, but it

may indicate some sort of passivation or poisoning of the $CeO_2$ surface by polyimide and its combustion products.

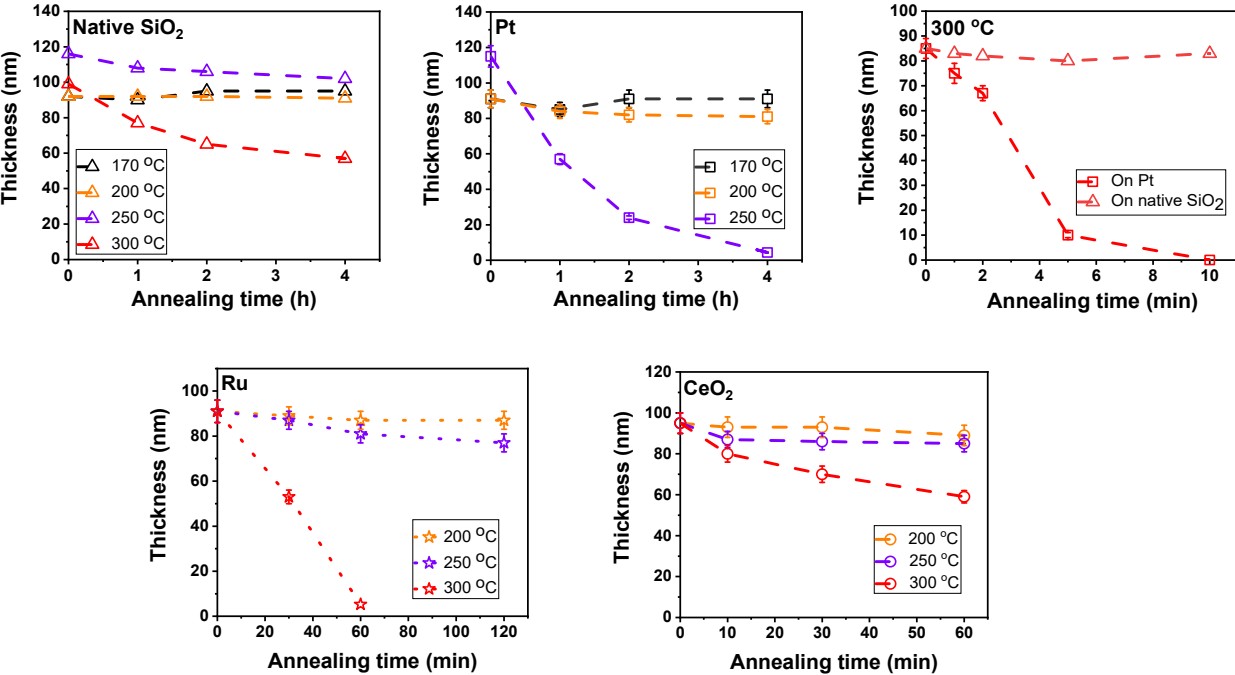

**Figure 2.** Polyimide film thicknesses on different surfaces versus annealing time in air.

As compared to normal etching processes, in this novel process, etching occurs oppositely, upward from the bottom of the polymer film toward its top surface. The overall process can be divided into three main steps: diffusion of $O_2$ through the polymer to its interface with the underlying material, catalysed reaction of $O_2$ with the polymer at the interface, and diffusion of volatile reaction products to the surface and away from the polymer. It is reasonable to assume that the polymer films were about the same quality on the different surfaces. Therefore, diffusion processes cannot explain the observed large differences in the etching rates, and these differences must arise from the different reaction rates caused by the catalytic effects of the underlying materials.

### 3.2. Self-Aligned Etching and Deposition on Patterned Pt/SiO₂ Surface

Self-aligned area-selective etching of polyimide thin films from Pt versus native $SiO_2$ was studied and successfully demonstrated on a patterned Pt/native $SiO_2$ surface with the feature sizes in the micrometre scale. Polyimide thin film with a thickness of about 90 nm was deposited on the patterned surface by MLD. After the polyimide deposition, the sample was annealed at 300 °C for 30 min under ambient atmosphere. While in Figure 2, 10 min annealing at 300 °C was found to be sufficient for removing such a polyimide film, here, a longer annealing time of 30 min was used to ensure complete polymer removal, as any residues left on the Pt surface might complicate the following ALD process. After the annealing, the sample was examined by cross-sectional scanning electron microscopy (SEM). The SEM images (Figure 3a) show that the polyimide film was completely removed from the Pt surface, while on the native $SiO_2$ surface, 80 nm of the original 90 nm polyimide film remained. On the $SiO_2$ surfaces next to the catalytic Pt regions, a larger thickness decrease of around 40 nm was observed, which might be caused by the diffusion of oxygen atoms from the Pt surface to the nearby $SiO_2$ surface. Alternatively, and even more likely as evidenced below, the thinning may be due to Pt residues that deposited on $SiO_2$ because the patterning was conducted by evaporation through a shadow mask. Regardless, the polyimide extended all the way to the edge of the Pt region (Figure 3c),

and the remaining 50 nm of polyimide on the regions next to Pt were more than enough to make the process self-aligned.

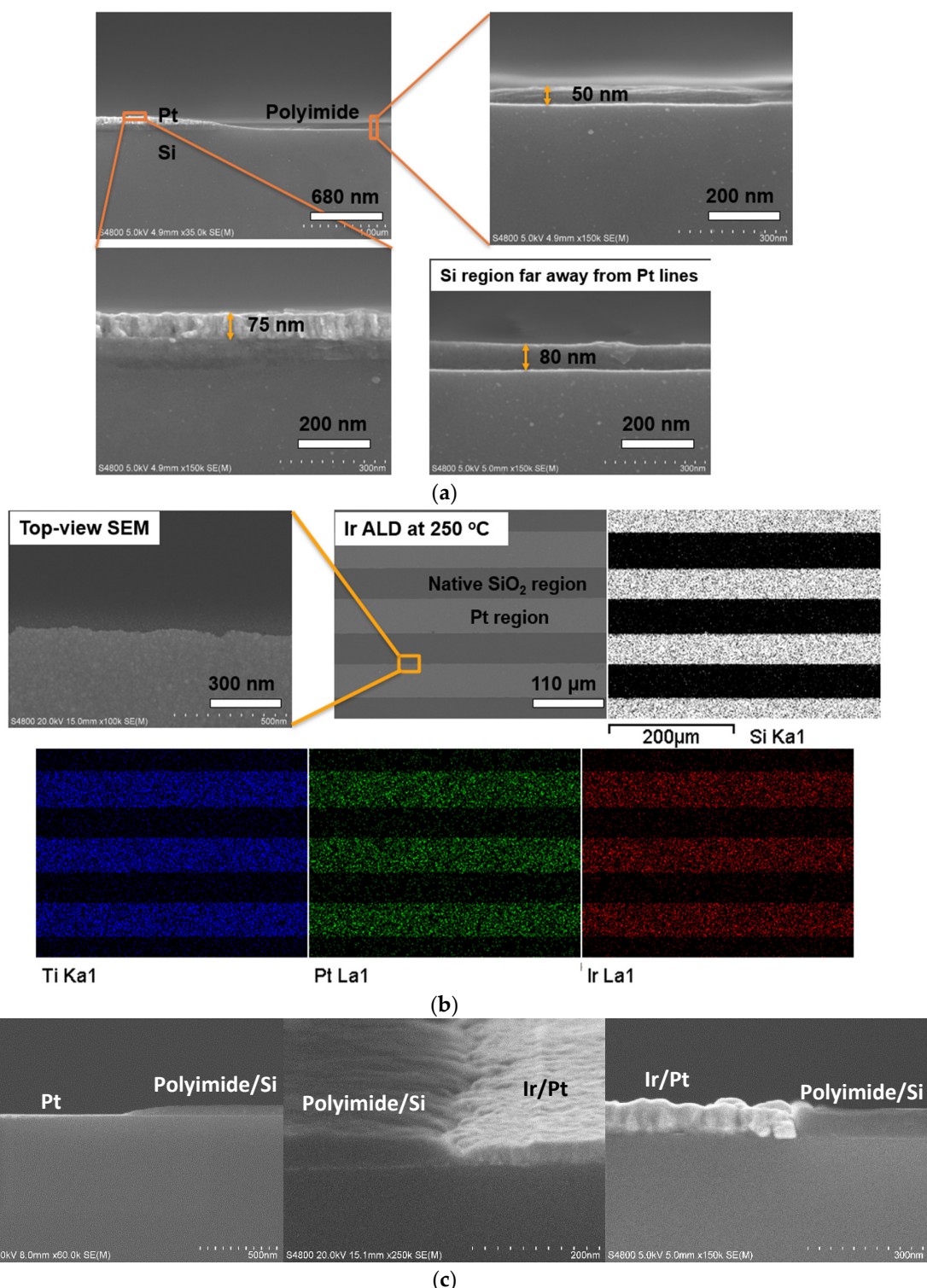

**Figure 3.** (**a**) Cross-sectional SEM images showing the success of area-selective etching of polyimide films on a patterned Pt/SiO$_2$ sample. (**b**) Top-view SEM images and the corresponding EDX maps demonstrating the success of the subsequent area-selective ALD of Ir on Pt. (**c**) SEM images of the SiO$_2$–Pt boundary after the polyimide etching (**left**) and Ir deposition (**middle**). The image on the (**right**) shows a cross section of the SiO$_2$–Pt boundary from a sample wherein the line was patterned with a lift-off process and then processed similarly.

As illustrated in Figure 1, area-selective etching of polymers can be followed by area-selective thin film deposition, wherein the patterned polymer thin film is used as an inhibition layer to prevent film growth. This allows simple and robust self-aligned deposition of metal-on-metal. An ALD iridium process was chosen for this demonstration. Ir films were deposited from $Ir(acac)_3$ and $O_2$ at 250 °C [23]. After 1000 ALD cycles, the sample was characterized by SEM and EDX (energy dispersive X-ray spectroscopy). As evidenced by the top-view SEM image (Figure 3b), no Ir growth was observed on the polyimide-covered $SiO_2$ surfaces. It can therefore be concluded that the polyimide films remaining on the native $SiO_2$ surfaces successfully prevented the ALD Ir growth. The EDX maps in Figure 3b show the distributions of Ir, Pt, Si, and Ti. The Ti signal came from a Ti adhesion layer that was underneath the Pt layer, whereas the Si signal is blocked by the metal layers in these areas. The Ir signal was well aligned with the Pt signal, thereby proving that Ir was selectively deposited on the Pt surfaces while no growth occurred on the polyimide-covered native $SiO_2$ surfaces, which further confirms the SEM result. As evaluated from the EDX intensities [25], the thickness of the Ir film grown on the Pt surface of the pattern was around 35 nm, which is consistent with the growth rate reported in the literature [23]. This also verifies that the polyimide etching was successful and complete, because residues would have disturbed the sensitive ALD Ir process.

The edge between the metal and polyimide regions was rough because the catalytic Pt line was created by evaporating through a shadow mask, which unavoidably produced line edge roughness and some isolated Pt islands on $SiO_2$ close to the Pt line. For the same reason, it is difficult to examine the sharpness of the boundary down to nanometre accuracy, but the cross-sectional SEM images in Figure 3c on the line edge after the etching and after the Ir deposition give a promising indication. As a further test, a better-defined Pt line edge was created by a lift-off method, and the overall process of polyimide deposition, area-selective etching and area-selective Ir deposition was repeated. In this case, the selectively deposited Ir indeed formed a vertical boundary against the polyimide on $SiO_2$ that retained its thickness all the way to the Pt line edge (Figure 3c). This also supports that the polyimide thinning observed next to the Pt line in Figure 3a was due to Pt residues on the $SiO_2$ area. Further research is needed to verify how the process scales to smaller features.

### 3.3. Catalytic Combustion of PMMA in $O_2$ Atmosphere

The catalytic combustion mechanism makes the novel self-aligned patterning method applicable for any polymer thin film that is thermally stable at the corresponding temperatures. Therefore, the etching of another polymer, PMMA, was studied similarly to that of polyimide. PMMA is an important and widely used resist material in semiconductor processing. PMMA thin films with thicknesses of 90–130 nm were prepared by spin-coating. The PMMA thin films were found easier to combust away than the polyimide thin films. The combustion of the PMMA thin films on the native $SiO_2$, again considered as a noncatalytic surface, started at 300 °C (Figure 4), which was the same onset temperature as for the polyimide combustion on $SiO_2$ but the etching of PMMA (Figure 4) proceeded faster than that of polyimide (Figure 2). Even so, a thickness decrease of only 40 nm was measured for the PMMA thin films after 1 h annealing, showing that the PMMA combustion reaction was still reasonably slow on the native $SiO_2$. At lower temperatures, no noticeable thickness decrease in PMMA films was measured after 1 h annealing (Figure 4).

Again, the Pt surface exhibited remarkable catalytic activity in the combustion and etching of PMMA. As seen in Figure 4, on Pt, the PMMA combustion reaction started at only 200 °C, at which a 100 nm thick PMMA film was completely removed within an annealing time of only 10 min, showing an impressive combustion rate and selectivity of PMMA against $SiO_2$. The PMMA combustion rates were apparently even higher at annealing temperatures of 250 and 300 °C, but they did not show up in our experiments because of the limited thickness of the PMMA films used, only 90–100 nm. On Ru, the PMMA combustion became prominent at 250 °C; a PMMA thin film with a thickness of 100 nm was completely removed after 60 min annealing. At a higher annealing temperature of

300 °C, the combustion was accelerated, and 10 min was enough for the complete removal of a 100 nm PMMA film.

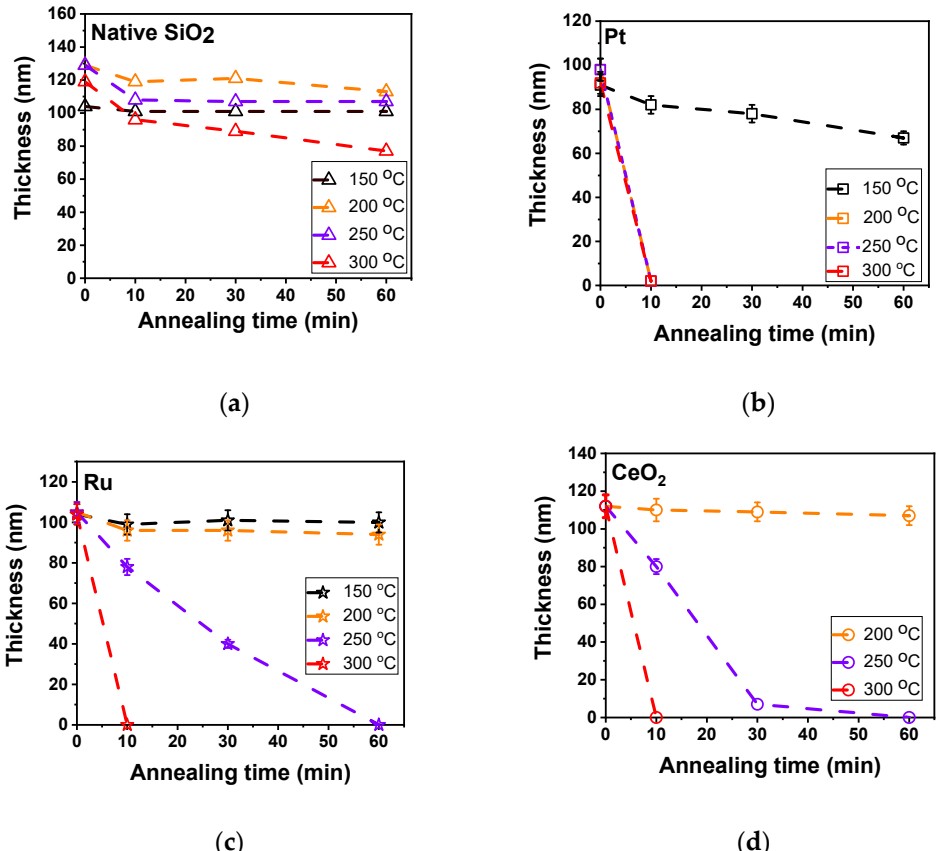

**Figure 4.** PMMA film thicknesses on different surfaces versus annealing time in air (**a**) Native SiO$_2$; (**b**) Pt; (**c**) Ru; (**d**) CeO$_2$.

While CeO$_2$ showed no catalytic effect on polyimide combustion or etching (Figure 2), the CeO$_2$ surface did exhibit impressive catalytic activity in the PMMA combustion reaction, as seen in Figure 4. The etching of PMMA on CeO$_2$ started at 250 °C with a rate even higher than that on the noble metal Ru. After further increasing the annealing temperature to 300 °C, the PMMA etching rate was greatly increased, such that an around 110 nm-thick PMMA film was completely removed within only 10 min. This is an important result because it adds an insulator to the ranks of catalytic materials that can be used for self-aligned etching processes.

The outstanding catalytic performance of Pt, Ru, and CeO$_2$ in combustive PMMA etching strongly supports the hypothesis that the novel approach of self-aligned area-selective etching of polymers relying on the catalytic nature of the underlying materials can be extended to other polymers as well.

### 3.4. Catalytic Etching of PMMA in H$_2$ Atmosphere

The self-aligned etching of polymers with O$_2$ has one potential drawback: the process may undesirably oxidize some materials, particularly metals and metal nitrides, on the surface. Therefore, the question of whether O$_2$ could be replaced with H$_2$ as an etchant gas, thereby changing the oxidizing conditions to reducing ones, was explored. The main idea is still the same: H$_2$ molecules diffuse through the polymer and become catalytically activated by the underlying material that dissociates them into atomic hydrogen. The hydrogen atoms then attack and cleave the carbon-carbon bonds of the polymer, which leads into its etching as small volatile molecules. Such hydrogenolysis reactions have been

extensively studied for the fracturing of saturated hydrocarbons [26] and polymers [27] and are known to be significantly enhanced by catalysts such as noble and transition metals.

Here, six metal surfaces that have potential for catalytic hydrogenolysis were studied, including noble metals (Pt and Pd) and transition metals (Cu, Co, W, and Ti). A native $SiO_2$ surface was studied as the noncatalytic surface for comparison. PMMA thin films with thicknesses of 80–120 nm were prepared by spin-coating. Forming gas was used as the $H_2$ atmosphere, and the annealing temperatures were 250–350 °C. The samples were enclosed into a tubular vacuum oven that was carefully purged by $N_2$ to remove air inside and then heated to the selected temperatures, at which it was kept for 30 min. As shown in Figure 5, among the studied annealing temperatures, more polymer was removed from the metal surfaces than from the native $SiO_2$, demonstrating a catalytic effect of these metals on PMMA removal. Notably, 300 °C was identified as an optimal temperature that enabled the greatest difference in polymer removal between the catalytic surfaces and the reference $SiO_2$ surface. At this temperature, Cu and Ti showed the best catalytic performance (Figure 5). The polymer was essentially completely removed from Cu and Ti, with only 2 nm remaining on Cu according to an ellipsometer, whereas almost 100 nm remained on the native $SiO_2$. Completion of the removal from Cu may require an additional plasma etching step, but the difference in thicknesses gives a good margin to ensure self-aligned etching of PMMA from Cu and Ti over $SiO_2$, for example. At the temperatures of 250 and 350 °C, the differences in the etch rates between the catalytic surfaces and $SiO_2$ were not as significant as that at 300 °C.

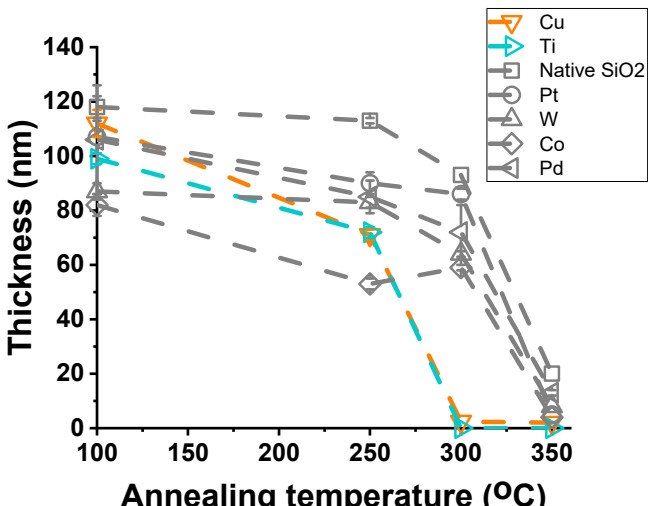

**Figure 5.** Thicknesses of PMMA films remaining on different surfaces after annealing for 30 min under $H_2$. Curves for the most catalytic surfaces (Cu and Ti) are shown in colours, while those for inactive surfaces and surfaces comparable to $SiO_2$ are in grey.

*3.5. Catalytic Thermal Cracking of Polyimide in Inert Atmosphere*

Polymers can also be decomposed by thermal cracking reactions without any reactant molecules. Thermal cracking is thought to be one of the most cost-efficient methods to produce fuels from polymer wastes [28]. Thermal cracking of polyethylene, polypropylene, and polystyrene has been extensively studied because of the high quality of cracking products [29]. Catalysts can significantly accelerate also the cracking reactions [30]. Porous zeolites and their derivatives have shown high catalytic performance due to their strong surface acidity. Here, the catalytic cracking of polyimide films was studied on various metal surfaces, including Ru, Co, W, Pt, Ti and Cu, with native $SiO_2$ as a reference.

Polyimide thin films with thicknesses of 100–150 nm were prepared by MLD. $N_2$ gas was used as the inert gas atmosphere. The annealing temperatures were 350–450 °C, and the annealing time was 30 min, in all the experiments. Comparing the polyimide

thicknesses on the different surfaces (Figure 6), Cu and Ru showed the highest catalytic performance among the surfaces examined. The optimal temperature enabling the largest difference in polyimide removal between the catalytic surfaces and the native $SiO_2$ was 400 °C. At this temperature, the polyimide thickness decreased to about 20 nm on Ru and Cu, whereas on other surfaces, the thickness decrease was much smaller and comparable to that on $SiO_2$. On Pt, the behaviour was different; there was clear etching at 350 °C, but as the temperature rose to 400 °C, the etching rate increased more slowly than that on Cu and Ru. Therefore, the highest selectivity was observed between the catalytic Cu and Ru surfaces against the noncatalytic $SiO_2$, Co and W. However, as compared with the combustion and hydrogenolysis etching reactions, the cracking reactions required higher temperatures with a consequent increased risk of damage to the thin-film structures. Also, a thicker residue layer remained, which is understandable considering that there was no other source of hydrogen and oxygen for forming volatile etch product molecules besides the polyimide itself. Because etching by cracking reactions requires the highest temperatures and must be complemented with some other etch method, it is clearly the least favourable of the three alternatives studied here.

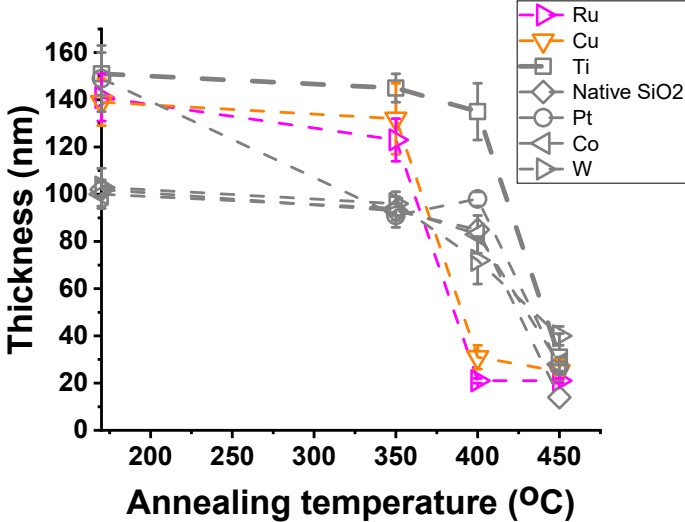

**Figure 6.** Thicknesses of polyimide films remaining on different surfaces after annealing for 30 min under $N_2$ atmosphere. Curves for the most catalytic surfaces (Ru and Cu) are shown in colours, while those for inactive surfaces and surfaces comparable to $SiO_2$ are in grey.

## 4. Conclusions

In summary, a novel self-aligned thin-film patterning approach was successfully demonstrated. By utilizing the catalytic effect of the underlying materials on the polymer etching reactions, area-selective removal of polymers (PMMA and polyimide) from the catalytic surfaces (noble metals, transition metals, and dielectric $CeO_2$) versus the noncatalytic surfaces ($SiO_2$) was achieved. In addition, area-selective thin film deposition was performed with the patterned polymer thin film as a growth-inhibition layer, as proven by the area-selective ALD of Ir on Pt versus polyimide covered $SiO_2$. Because of its self-aligned nature and potential applicability to a wide range of polymers, as well as the flexibility it provides in choosing different processing atmospheres (oxidative, reductive, or inert gas atmosphere), thin-film patterning through area-selective etching of polymers is expected to open many new possibilities in the semiconductor industry and beyond. By demonstrating and verifying the concept of self-aligned etching, the present study also opens a plethora of research questions for the following studies: with which combinations of polymer–catalyst surface–etchant gas–temperature selectivity can be achieved, how the process scales to structures with the smallest dimensions, how to convert a noncatalytic surface to a catalytic surface, and so on. The final question will be where in the overall IC fabrication process flow self-aligned etching can be used. In this respect, an important

parameter, besides the etching gas compatibility, is thermal budget (temperature and time). The maximum processing temperature allowed varies throughout the fabrication process, being the highest on the silicon level (front-end) while making transistors and decreasing toward the completion of interconnects (back-end). The most efficient self-aligned etching processes demonstrated here were completed in minutes at 200 °C, which should meet the tightest thermal budget requirements, whereas the compatibility of 300 °C processes depends largely on the materials involved and requires careful study.

**Author Contributions:** All authors contributed to the conceptualization of the project, the planning of experiments, and the interpretation of results. C.Z. conducted the experiments under the supervision of M.L. and M.R., C.Z. and M.R. wrote the first version of the paper, and all authors contributed to finalizing the paper. All authors have read and agreed to the published version of the manuscript.

**Funding:** Financial support from ASM Microchemistry, the Academy of Finland (ALD of Noble Metals and Their Compounds, decision number 309552), and the China Scholarship Council (File No. 201507040043) are gratefully acknowledged.

**Institutional Review Board Statement:** Not applicable.

**Informed Consent Statement:** Not applicable.

**Data Availability Statement:** Not applicable.

**Acknowledgments:** The authors thank Marko Vehkamäki for preparing metal surfaces by electron beam evaporation and Marianna Kemell for SEM characterization. The work was done in the ALD centre Finland research infrastructure.

**Conflicts of Interest:** The authors declare no conflict of interest.

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
