# Peer review of "Self-Aligned Thin-Film Patterning by Area-Selective Etching of Polymers"

_coatings, doi:10.3390/coatings11091124_

Round 1

Reviewer 1 Report

The paper proposes a new interesting approach to the etching of polymer materials. The manuscript is well written and the results are clearly presented. The paper can be published after a minor revision.

Minor questions/remarks:

1. On page 6 last paragraph it is stated that the etch occurs at the interface between the underlying catalytic material and the polymer film. During the bottom-up etching the contact between the catalyst and the polymer may be lost due to the polymer consumption (especially in patterned samples). Do the Authors consider the possibility of transport of active species from the catalytic surface to the polymer through an "air-gap" that might be formed in the course of the etch process?

2. What is the source of the large scatter in the initial film thickness in Fig 5 and Fig 6?

3. Fig5 and Fig 6 should be formatted.

Author Response

Thank you  for the positive feedback.

1. On page 6 last paragraph it is stated that the etch occurs at the interface between the underlying catalytic material and the polymer film. During the bottom-up etching the contact between the catalyst and the polymer may be lost due to the polymer consumption (especially in patterned samples). Do the Authors consider the possibility of transport of active species from the catalytic surface to the polymer through an "air-gap" that might be formed in the course of the etch process?                                                                                                                                                                                                        This is in principle possible but in such a case no such sharp interface as seen in Fig. 3 should form. Also, it is most reasonable to assume that atomic oxygen or hydrogen are the reactive species, and it is less clear if they could spread or spill over in atomic form. And because catalytic etching occurs also without etching gases, it looks like that air gap is not forming but the the film is collapsing downwards. The best we can do is to show images in Fig. 3 about the quality of the interface.

2.  What is the source of the large scatter in the initial film thickness in Fig 5 and Fig 6?

The scatter (about 20 %) in initial thicknesses on different materials derives from different growth on these materials. It does not affect the conclusions about etching, however.

3. Fig5 and Fig 6 should be formatted.

We agree the figures are at first sight quite hard to read, but our logic has been in emphasising with color those surfaces that were the most active. This explanation has been added to figure captions and with that in mind reading the figures should be straightforward.

Reviewer 2 Report

The Author described studies concerning on self-aligned thin-film pattering by selective etching of polymers.  Below, several aspects  have mentioned, which should be corrected and some doubts should be explained.

  1. The Abstract should be improved and contain the most important results from the manuscript.
  2. I suggest to use different colors in Fig. 6 in order to clarify picture.

Generally, the Author did some work. However it could not be published in present form in Coatings. I recommend deep and careful revision.

Author Response 

1. The Abstract should be improved and contain the most important results from the manuscript.

Thank you for pointing out that the abstract was lacking the main results. These have now been added.

2.  I suggest to use different colors in Fig. 6 in order to clarify picture.

Generally, the Author did some work. However it could not be published in present form in Coatings. I recommend deep and careful revision.

We agree the figure is at first look quite hard to read, but our logic has been in emphasising with color those surfaces that were the most active. This explanation has been added to figure captions and with that in mind reading the figure should be straightforward.

What comes to deep and careful revision, without further guidelines it is hard to see what would be expected. Also, the other reviewers seemed to be satisfied with the content.  This is the first report on area-selective etching and we note in conclusions that there indeed remains a lot to be done in the future: "By demonstrating and verifying the concept of self-aligned etching, the present study also opens up a plethora of research questions for the following studies: with which combinations of polymer - catalyst surface - etchant gas – temperature selectivity can be achieved, how the process scales to structures with the smallest dimensions, how to convert a non-catalytic surface to a catalytic surface, and so on. Final question will be where in the overall IC fabrication process flow the self-aligned etching can be used."

Reviewer 3 Report

The manuscript by Zhang et al describes the new approach in thin-film patterning by area-selective etching of polymers. Authors provided the mechanisms of etching and demonstrated the opportunity of applying this methodology for film patterning. In my opinion, the manuscript is well prepared and may me accepted for publication.

Author Response

Thank you very much for the very positive report.